# Scalable and Loosely-Coupled Multimodal Deep Learning for Breast Cancer Subtyping

**Mohammed Amer**                                    MOHAMMED.AMER@FUJITSU.COM
**Mohamed A. Suliman**                               MOHAMED.SULIMAN@FUJITSU.COM
**Tu Bui**                                           TU.BUI@FUJITSU.COM
**Nuria Garcia**                                     NURIA.GARCIA.UK@FUJITSU.COM
**Serban Georgescu**                                 SERBAN.GEORGESCU@FUJITSU.COM
*Fujitsu Research of Europe Ltd*
*Slough, United Kingdom*

**Editor:**

## Abstract

Healthcare applications are inherently multimodal, benefiting greatly from the integration of diverse data sources. However, the modalities available in clinical settings can vary across different locations and patients. A key area that stands to gain from multimodal integration is breast cancer molecular subtyping, an important clinical task that can facilitate personalized treatment and improve patient prognosis. In this work, we propose a scalable and loosely-coupled multimodal framework that seamlessly integrates data from various modalities, including copy number variation (CNV), clinical records, and histopathology images, to enhance breast cancer subtyping. While our primary focus is on breast cancer, our framework is designed to easily accommodate additional modalities, offering the flexibility to scale up or down with minimal overhead without requiring re-training of existing modalities, making it applicable to other types of cancers as well. We introduce a dual-based representation for whole slide images (WSIs), combining traditional image-based and graph-based WSI representations. This novel dual approach results in significant performance improvements. Moreover, we present a new multimodal fusion strategy, demonstrating its ability to enhance performance across a range of multimodal conditions. Our comprehensive results show that integrating our dual-based WSI representation with CNV and clinical health records, along with our pipeline and fusion strategy, outperforms state-of-the-art methods in breast cancer subtyping.

**Keywords:** Multimodal; Machine Learning; Cancer subtyping; Graph AI.

## 1 Introduction

Cancers are genetically and molecularly diverse (Dagogo-Jack and Shaw, 2018), making accurate subtype classification essential for personalized treatment (Wolf et al., 2022; Yin et al., 2020). This task involves integrating complex data sources like copy number variation (CNV), histopathology images, and electronic health records (EHR), which poses analytical challenges. Automating breast cancer subtyping can significantly improve clinical outcomes.

Multimodal ML addresses this by combining information across modalities using early, intermediate, or late fusion strategies (Steyaert et al., 2023). However, variability in data availability across tasks calls for flexible, scalable approaches. In response to these challenges,

we propose a loosely coupled multimodal framework that accommodates CNV, EHR, and whole slide image (WSI) data for PAM50 breast cancer subtyping (Parker et al., 2009). To balance efficiency and resolution, we combine image-based and graph-based WSI representations (Levy et al., 2020; Pati et al., 2020; Adnan et al., 2020). Our weighted logits late fusion method enables dynamic modality integration and outperforms SOTA approaches.

Our main contributions are:

- We present a scalable and flexible multimodal pipeline that can scale to large number of modalities and adapt to modality changes efficiently. The application of our method to breast cancer subtyping achieves better than SOTA performance and contributes to enhancing the clinical process.

- We propose a new late fusion strategy that shows significant improvement over the SOTA in different multimodal combinations.

- We develop a comprehensive pipeline that transforms WSI representations into graphs, enabling an augmented dual representation that leverages both WSI images and WSI-based graphs. This approach enhances model performance by combining local and global features, achieving state-of-the-art results on WSI-related tasks and multimodal integration

## 2 Related Work

Multimodal integration plays a crucial role in addressing various healthcare challenges by leveraging multiple patient data types to enhance decision-making. In recent years, several studies have demonstrated the effectiveness of multimodal approaches in disease classification and prognosis. For instance, Wang et al. (2021) successfully classify lung cancer subtypes using CNV and WSI data from TCGA-LUAD and TCGA-LUSC, extracting WSI features via a pre-trained InceptionV3 model with distance-weighted pooling. Similarly, Liu et al. (2022b) integrate CNV, gene expression, and WSI data from TCGA-BRCA for PAM50 breast cancer subtyping. Their approach employs PCA for genomic feature reduction, VGG16 for WSI feature extraction, and weighted averaging optimized via simulated annealing for final predictions.

Several multimodal methods use graph-based representations to model complex biological processes. For example, Ansarifar et al. (2022) integrate gene graphs from genome-wide association studies (GWAS) with brain graphs from resting-state fMRI to predict phenotypic outcomes, using graph convolutional networks (GCNs) for multimodal fusion. In a similar vein, Liu et al. (2022a) combine genetic, EHR, and MRI data to predict mild cognitive impairment, constructing multiple graphs from genomics and MRI features, applying GCNs for per-graph predictions, and aggregating the results using majority voting.

In our work, graph representations for WSIs are crucial. Bilgin et al. (2007) segmented WSIs to extract individual cells and modeled tissue structures as graphs based on spatial arrangements of cells. Lu et al. (2018) introduced Feature-Driven Local Cell Graphs (FeDeG), which construct cell graphs by considering spatial proximity and nuclear attributes like shape and size. Lu et al. (2020) further refined this approach by segmenting and classifying nuclei, clustering nearby nuclei into nodes, and defining graph edges based on a maximum distance connectivity threshold between cluster centers.

Fusion strategies combine multiple model predictions into a single, unified prediction. Tang et al. (2024) present a benchmark for evaluating deep model fusion techniques across various tasks, such as image and text classification, and text-to-text generation. Their comprehensive study explores fusion strategies like simple ensemble, weighted ensemble, and max-model predictor. In meta-learning (Huang et al., 2020; Stahlschmidt et al., 2022), fusion is achieved by training a separate model to optimize the combination of single-modality predictions. Intermediate fusion (Steyaert et al., 2023) integrates information at the intermediate model representation level, allowing cross-modal interactions during feature extraction.

Building on these foundations, we propose a flexible, loosely coupled multimodal integration method that scales efficiently to multiple modalities. Our fusion strategy is benchmarked against several SOTA fusion methods on breast cancer subtyping, demonstrating superior performance across different modality combinations. Furthermore, our findings reveal that the dual representation of WSIs as both images and graphs enhances performance in breast cancer subtyping and contributes to improving and automating the clinical application.

## 3 Method

We integrate CNV, WSI (image and graph), and EHR using a multimodal approach. Each modality is pre-processed and modeled independently to extract representations and predictions. The extracted features are then combined via intermediate and late fusion, with a second training phase optimizing fusion for the downstream task. Our decoupled design allows flexible modality addition/removal and easy adaptation of fusion strategies.

### 3.1 Fusion strategies

Our late fusion strategy can be formulated as follows Fig. 1i (A). Let $o^{(i)} \in \mathbb{R}^C$ denote the output logits of model $i$, where $C$ is the output dimension. The multimodal prediction logits are then computed as:

$$o_j = \sum_{i=1}^{M} w_{ij} \cdot o_j^{(i)} + b_j \tag{1}$$

where $M$ is the number of modalities, $o_j^{(i)}$ is the $j$th element of the output from modality $i$, $o_j$ is the $j$th element of the final multimodal prediction logits, $w_{ij}$ are non-normalized trainable weights, and $b_j$ is a bias term. A task-specific non-linearity is then applied to these logits to obtain the final multimodal probabilities.

#### 3.1.1 EXPERIMENTAL BENCHMARKING

We benchmarked our fusion strategy against multiple SOTA fusion methods from the literature (Liu et al., 2022b; Tang et al., 2024; Wu et al., 2019; Huang et al., 2020; Stahlschmidt et al., 2022; Steyaert et al., 2023). Additionally, we compared our approach with a Transformer-based fusion strategy. In this comparative method, each intermediate representation from a single-modality model is treated as a token in the Transformer's input sequence. A classification token is appended to this sequence, and an MLP is applied to the corresponding

output to generate the final multimodal prediction. Our Transformer encoder consists of six encoder layers, each with a dimensionality of 512 and multi-head attention with eight heads.

We use the following abbreviations to refer to SOTA fusion methods in the result tables. WE: Weighted Ensemble(Liu et al., 2022b; Tang et al., 2024), SE: Simple Ensemble (Tang et al., 2024), MP: Max-Model Predictor(Wu et al., 2019), ML: Meta-Learning(Huang et al., 2020; Stahlschmidt et al., 2022), IF: Intermediate Fusion(Steyaert et al., 2023), T: Transformer, WL: Weighted Logits (Ours), WLB: Weighted Logits with Bias (Ours).

## 3.2 WSI

### 3.2.1 Image Representation

WSIs, typically multi-gigapixel representations of tissue samples, present significant challenges for machine learning. Their large size and resolution make it infeasible to process the whole WSI in a single pass due to memory and storage limitations. Additionally, a substantial portion of the WSI often consists of background, which provides no meaningful information for the learning process.

Following previous work (Lu et al., 2021), our WSI pipeline starts by dividing the WSI into non-overlapping patches. After downsampling each patch, we segment the tissue from the background by 1) converting the patch into hue-saturation-value (HSV) color space, 2) applying median blurring to the saturation channel using a kernel size of 7 and 3) generating a tissue mask by thresholding the saturation channel using a binary threshold of 20 (for $0-255$ pixel values). We accept the patch only if the area of the tissue content is more than 5% based on the generated mask.

Our WSI model uses a classification head atop a pre-trained backbone (Inceptionv3 (Szegedy et al., 2016), VGG16 (Simonyan and Zisserman, 2014), or Dinov2 (Oquab et al., 2023)). Each patient has multiple WSIs and patches, with patient-level labels requiring pooling of patch predictions.

We evaluated output-level pooling (majority vote, mean logits and mean probabilities) and intermediate-level pooling (mean or distance-weighted average (Wang et al., 2021)) Fig. 1i (B). The model is trained with weak supervision, where patches inherit patient labels.

Two pooling strategies were tested: (1) patch-wise training, freezing the backbone, then pooling intermediate features for final classification; (2) end-to-end training with pooled patch features. For efficiency, we select the top 50 patches per patient that contain the most tissue.

For Inceptionv3, VGG16 and Dinov2, the input patches are further pre-processed by: 1) resizing into $342 \times 342$ for Inceptionv3 or $256 \times 256$ for VGG16 and Dinov2 using bilinear interpolation, 2) applying a central crop of $299 \times 299$ for Inceptionv3 or $224 \times 224$ for VGG16 and Dinov2 and 3) rescaling the pixel values to $[0.0, 1.0]$ and normalizing using mean=$[0.485, 0.456, 0.406]$ and std=$[0.229, 0.224, 0.225]$.

### 3.2.2 Graph Representation

The WSI-to-graph pipeline begins by extracting tissue regions and removing large blank areas. We use a model combining ResNet-50 (encoder), U-Net decoder, and an F-CNN

for pixel-wise classification, pre-trained on the BCSS dataset (Amgad et al., 2019). Post-processing includes removing mask objects $< 10^4$ pixels, morphological closing, and removing holes $< 10^3$ pixels. An example is shown in Fig. 1ii.

After background suppression, we construct a graph $\mathcal{G}(W) = (\mathcal{V}, \mathcal{E})$ for each WSI $W$, where nodes $\mathcal{V}$ represent patch-derived features $\mathbf{X} = (\mathbf{x}_{v_i})$, following Graham et al. (2019); Lu et al. (2022). WSIs are divided into $512 \times 512$ patches. Each patch is processed by Hover-Net (Gamper et al., 2020), pre-trained on PanNuke, to segment and classify nuclei into five types. Using SlideGraph+ (Lu et al., 2022), we cluster patches based on spatial proximity and nuclear composition via a similarity kernel (0: similar, 1: dissimilar), with a spatial threshold of 2000. Pairwise similarities are computed using $e^{-\gamma \|v_i - v_j\|}$ with $\gamma = 0.001$, and clusters are formed via average linkage with a 0.8 cut-off. Features and coordinates within each cluster are averaged to form nodes. Edges $\mathcal{E}$ are added using Delaunay triangulation with a 4000 distance threshold.

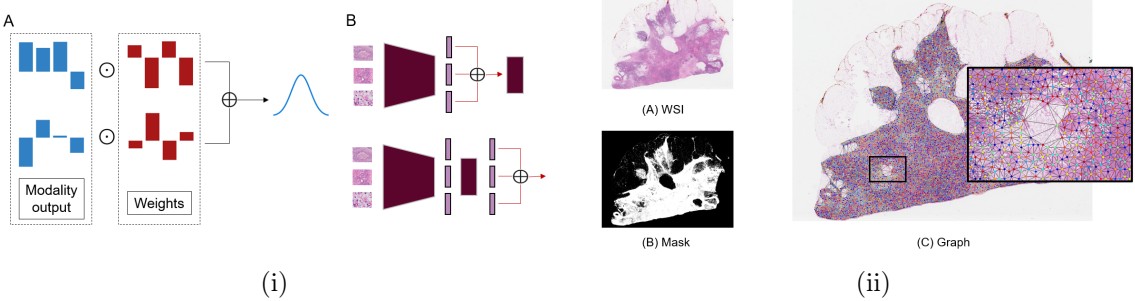

(i)             (ii)

Figure 1: (i) The model structure. (ii) Graph generation steps.

The final WSI graph is then passed through a graph neural network model with $L \geq 2$ layers. The input layer (i.e., $L = 0$) is set to have the following structure: linear layer + Batch Normalization + non-linear activation function. This is then followed by a linear layer at $L = 1$. Following that, we set $L - 1$ graph convolution layers based on the PNA graph convolution architecture (Corso et al., 2020). The output from each layer $L \geq 1$ is passed through a linear classification layer with $K$ outputs, where $K$ is the number of desired classes. The output of each classification layer at stage $l$ is being added to one at the previous layer $l - 1$. The intermediate representation is generated by pooling the outputs of the last graph convolution layer. A hyper-parameter optimization process is conducted to identify the best $L$, layers dimensionality, activation function, learning rate (LR), and dropout probability ($p$). The best model is found to be at $L = 3$ with a dimensionality of $[32, 16, 8]$, lr $= 1 \times 10^{-4}$, $p = 0.2$, and a RELU activation function.

### 3.3 Genetic modality

CNV represents the gain or loss of DNA segments, capturing duplication and deletion events across the genome. Each gene is assigned an integer value $g_i \in \{-2, -1, 0, 1, 2\}$, where $-2$ indicates a homozygous deletion (complete loss of both copies), $-1$ represents a heterozygous deletion (loss of one copy), 0 corresponds to a diploid state (normal copy number), $+1$

denotes a single copy gain, and +2 signifies amplification (multiple extra copies). We pre-process CNV data by removing genes with missing values.

For CNV modeling, we employ a neural network architecture consisting of a backbone network followed by a classification head. The backbone network is responsible for encoding the CNV data into an intermediate representation, which is subsequently fed into the classification head to generate predictions. We explore different architectural choices for both components, experimenting with standard feedforward neural network, and Self-Normalizing Networks (SNNs) (Klambauer et al., 2017), a specialized architecture designed to improve training stability and robustness through self-normalizing properties.

### 3.4 EHR

EHR includes clinical history, lab tests, demographics, and other patient clinical information. We remove features with over 75% missing values and uninformative fields, then categorize features as numerical, ordinal, or categorical. After converting and imputing values (using k-NN), we one-hot encode categorical features and apply z-scoring.

We test two EHR classification methods. The first uses a vanilla MLP as a backbone for intermediate features and a classification head. The second converts EHR data into text key-value pairs and uses a Transformer encoder (fine-tuned BERT (Devlin et al., 2018) or RoBERTa (Liu et al., 2019)) with a classification head for prediction.

## 4 Experiments

We evaluate our method on PAM50 breast cancer subtyping (Parker et al., 2009) using WSI, CNV, and EHR data from TCGA-BRCA. WSIs are obtained from TCGA; CNV and clinical data are obtained from cBioPortal (Cerami et al., 2012); and PAM50 labels from Netanely et al. (2016). After excluding incomplete cases, a total of 977 patients remain.

The four subtypes—Luminal A (53.5%), Luminal B (20.6%), Basal-like (18.1%), and Her2-enriched (7.8%)—lead to class imbalance. To address this, we mitigate via oversampling, stratified sampling, or loss weighting.

Evaluation uses 10-fold cross-validation, reporting accuracy and macro-AUROC (average AUC across classes). CNV uses an SNN with 8192–2048 channel layers and oversampling. The WSI model uses Inceptionv3, with majority voting and weighted loss. For clinical data, it is modeled with an MLP (128–64 hidden layers). All models use the Adam optimizer with default settings.

## 5 Results and Discussion

In this study, we demonstrate the importance of leveraging multimodal data for healthcare applications in a scalable and flexible manner. To showcase this potential, we apply our methodology to PAM50 breast cancer subtyping, an important task in precision oncology for breast cancer. By integrating multiple data modalities, we aim to significantly enhance predictive performance compared to single-modality approaches.

Tables 1 to 3 presents the results of our extensive benchmarking experiments, where we evaluate both single-modality models and various multimodal combinations. Specifically, we compare our proposed method—weighted logits fusion, with and without a bias

Table 1: Accuracy and Macro-AUC comparison between single modality models for breast cancer subtyping.

| Modality | | Accuracy | Macro-AUC |
|---|---|---|---|
| CNV | | 70.25 | 0.8284 |
| WSI | Image | 66.96 | 0.8080 |
| WSI | Graph | 70.23 | 0.8350 |
| Clinical | | 70.43 | 0.8522 |

| CNV | WSI Image | WSI Graph | Clinical | WE | SE | MP | ML | IF | T | WL* | WLB* |
|---|---|---|---|---|---|---|---|---|---|---|---|
| | ✓ | ✓ | | 68.57 | 68.64 | 68.44 | 66.52 | 66.23 | 66.02 | 68.68 | **69.51** |
| ✓ | ✓ | | | 72.57 | 73.24 | 73.14 | 71.96 | 70.97 | 73.54 | **75.41** | 75.12 |
| ✓ | ✓ | ✓ | | 73.3 | 74.52 | 73.72 | 74.12 | 71.66 | 71.87 | **76.41** | 76.23 |
| ✓ | ✓ | | ✓ | 74.10 | 76.23 | 74.73 | 75.10 | 72.08 | 73.49 | **76.88** | 76.78 |
| ✓ | ✓ | ✓ | ✓ | 73.96 | 76.91 | 74.76 | 77.15 | 71.37 | 74.25 | 77.92 | **78.13** |

Table 2: Accuracy comparison between SOTA and our method for breast cancer subtyping. Our method is marked with '*'. Best performance is bolded.

term—against a Transformer-based fusion baseline and five SOTA fusion methods for both intermediate and late fusion.

Our findings highlight the substantial benefits of multimodal integration. Compared to single-modality models, our method significantly improves classification performance. The highest accuracy achieved by a single-modality model is 70.43%, whereas our multimodal approach, utilizing all available modalities, achieves 78.13%. Similarly, for macro-AUC, the best single-modality performance reaches 0.8522, while multimodal fusion boosts it to 0.9153.

Across different multimodal combinations, our approach consistently outperforms SOTA methods in terms of accuracy. Regarding macro-AUC, our method surpasses SOTA in nearly all cases, except for one combination—CNV, image, and clinical data—where it is the second best performance. The performance gains vary depending on the modality combination, with improvements ranging from minor enhancements to approximately 2%, as observed in conditions such as CNV+image+graph.

Our results further demonstrate the advantage of integrating both image and graph-based representations of WSIs. When the graph representation is incorporated into CNV+image and CNV+image+clinical configurations, our method yields an approximate 1% increase in accuracy and a 0.02 improvement in macro-AUC. These findings suggest that the graph representation effectively captures complementary information, enhancing the predictive capability of our fusion strategy.

We conduct an interpretability analysis to understand the key biological factors influencing our breast cancer subtyping models. For **structured data**, we apply Integrated Gradients (Sundararajan et al., 2017; Chen et al., 2022) to our CNV and Clinical models, ranking features based on attribution scores across test samples. Our analysis Fig. S2 (A) highlights biologically relevant genes linked to breast cancer, such as TIMM17A (Salhab et al., 2012; Xu et al., 2010; Yang et al., 2016), CRTC2 (Brown et al., 2009; Brown

| CNV | WSI | | Clinical | WE | SE | MP | ML | IF | T | WL[*] | WLB[*] |
| | Image | Graph | | | | | | | | | |
|---|---|---|---|---|---|---|---|---|---|---|---|
| | ✓ | ✓ | | 0.8465 | 0.8569 | 0.8506 | 0.8274 | 0.8128 | 0.8220 | **0.8616** | 0.8604 |
| ✓ | ✓ | | | 0.8728 | 0.8738 | 0.8610 | 0.8692 | 0.8432 | 0.8589 | 0.8835 | **0.8836** |
| ✓ | ✓ | ✓ | | 0.8736 | 0.8931 | 0.8730 | 0.8794 | 0.8349 | 0.8395 | **0.9000** | 0.8995 |
| ✓ | ✓ | | ✓ | 0.8965 | **0.9074** | 0.8873 | 0.8912 | 0.8429 | 0.8592 | 0.8976 | 0.9008 |
| ✓ | ✓ | ✓ | ✓ | 0.8978 | **0.9153** | 0.8889 | 0.9006 | 0.8369 | 0.8541 | **0.9153** | **0.9153** |

Table 3: Macro-AUC comparison between SOTA and our method for breast cancer subtyping. Our method is marked with '*'. Best performance is bolded.

and Simpson, 2010; Samarajeewa et al., 2013), and CD34(Chauhan et al., 2003; Chen et al., 2015; Cimpean et al., 2005), as well as crucial clinical features like HER2 IHC score, ER/PR Status, and Fraction Genome Altered, which are well-established factors in breast cancer prognosis and treatment decisions (Tsai et al., 2019; Zhou et al., 2019; Gehrig et al., 1999; Allison et al., 2020). For **unstructured data**, we interpret our WSI-derived graph model by generating heatmaps of node-level activations overlaid on the original WSI Fig. S2 (B). To improve global interpretability, we merge bounding boxes of top-contributing nodes, revealing important regions in the tissue. These regions are further analyzed for their cellular composition using HoverNet.

For **multimodal data**, we extract interpretability insights from the learned weights of our late fusion model Fig. S2 (C). We find that contributions from WSI-based CNN and graph models collectively account for more than half of the prediction scores, with the graph model having the highest attribution (28%) and the CNN model the lowest (23.75%). This highlights the complementary nature of image and graph-based WSI representations in multimodal fusion.

## 6 Conclusion

We present a scalable multimodal approach that adapts to varying medical data types, using a novel late fusion strategy that outperforms SOTA methods. By utilizing both image-based and graph-based WSI representations, we significantly enhance data utility without requiring additional acquisition. This dual representation substantially boosts breast cancer subtyping performance, supporting seamless integration into clinical workflows.

Our study demonstrates the considerable potential of our method for breast cancer subtyping, yet its applicability extends far beyond this domain. A promising future direction involves generalizing our approach to a broader range of medical fields that can benefit from multimodal integration, thereby advancing precision medicine and improving patient outcomes across diverse healthcare applications

## 7 Acknowledgement

The results published here are in whole or part based upon data generated by the TCGA Research Network: https://www.cancer.gov/tcga.

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

## Appendix A. Supplementary Material

Figure S1 presents a principal component analysis (PCA) of the output logits for single-modality models and their fusion using our weighted logits strategy. The visualization illustrates that weighted logits fusion leads to improved stratification of samples based on their corresponding class, providing insight into the observed performance improvements.

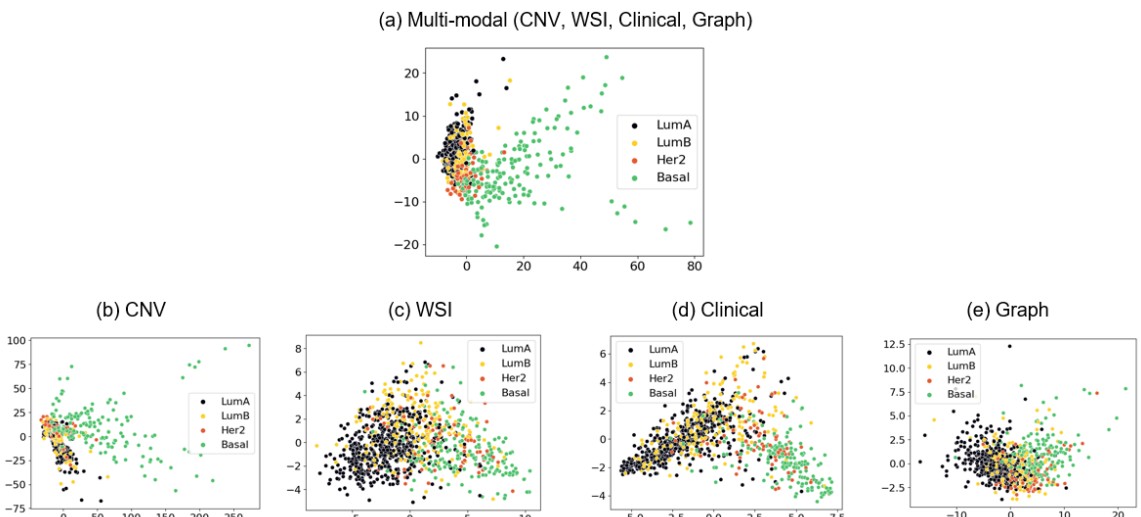

Figure S1: Dimensionality reduction using PCA for the fused logits using our weighted logits fusion strategy and the corresponding single-modality logits for breast cancer subtyping.

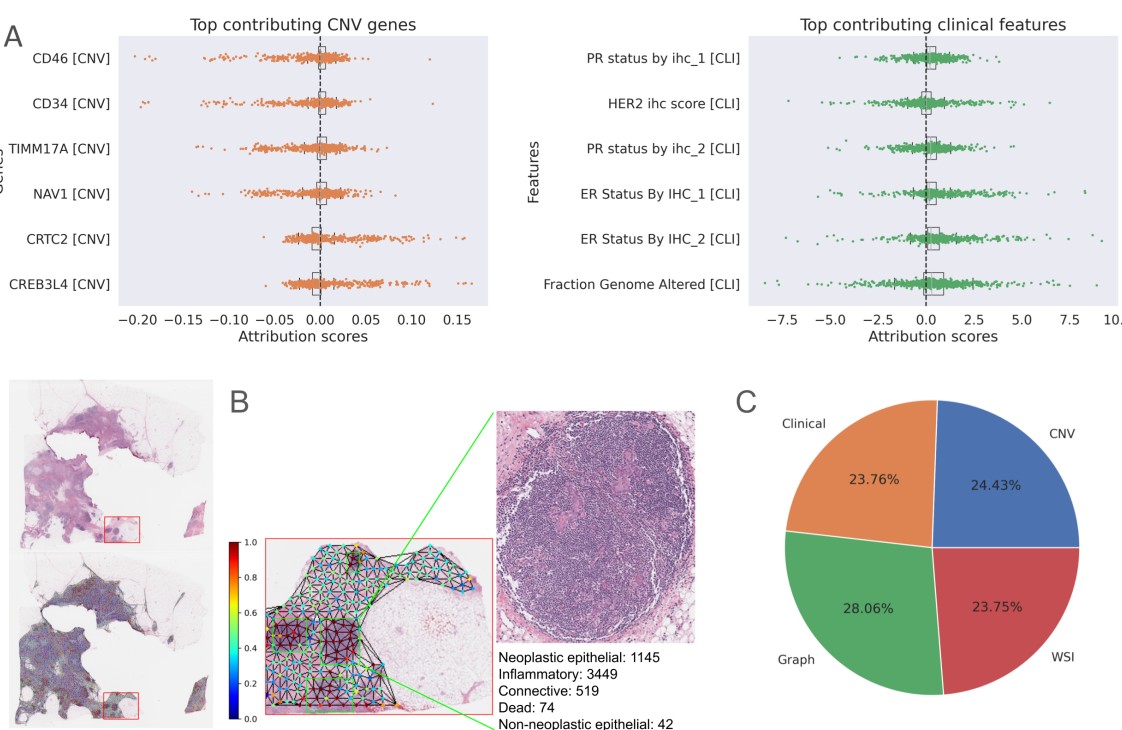

Figure S2: Interpretability analysis of our breast cancer subtyping models. A) Integrated Gradient scores for selected features from top 45 most contributing features on structural data. B) Graph interpretability for a representative WSI at node level and regional level. C) Modal contribution for our multimodal model.

