# OpenReview forum: "Scalable and Loosely-Coupled Multimodal Deep Learning for Breast Cancer Subtyping"
_MICCAI.org/2025/Workshop/COMPAYL — COMPAYL 2025_

### Official Review · Reviewer_ZzEV · 2025-07-11
**Scalable multimodal subtyping in breast cancer, promising with some clinical gaps**

**Rating:** 4
**Confidence:** 5

**Review:**

This paper presents a multimodal deep learning framework that integrates histopathology images (WSI), copy number variation (CNV), and electronic health records (EHR) to classify breast cancer into molecular subtypes (PAM50). The novelty lies in the combination of image-based and graph-based WSI representations and a flexible late fusion approach that allows modalities to be added or removed without retraining.
I value the focus on scalability and modality independence, which reflects real clinical conditions where not all data are consistently available. The integration of pathology with clinical and omics data supports the precision medicine vision central to this workshop. That said, some of the biological insights remain underexplored, and the clinical translation aspects could be better emphasized.

Strenghts:
* Highly relevant to workshop's goals: true integration of pathology, omics, and clinical data.
* The dual WSI representation (CNN + graph) is well-motivated and shows consistent performance gains.
* Loosely-coupled fusion and scalability

Weaknesses:
* Clinical translation is underdeveloped: the authors do not discuss how their model would be used by clinicians or integrated into decision-making.
* Graph representations of WSI are interesting, but the paper misses the opportunity to extract or explain tissue-level patterns (such as tumor–stroma interactions, immune cells).
* No external validation dataset. The results are all on TCGA, which limits generalizability.

---

### Official Review · Reviewer_qs2R · 2025-07-13
**Minor improvements with simple fusion strategy**

**Rating:** 3
**Confidence:** 4

**Review:**

**Summary**: The manuscript proposes a framework for multimodal integration of histopathology images, modeled by image representations or graph representations, with copy number variations, and clinical reports by using a weighted average as late fusion of modalities. The framework is evaluated on the application of molecular subtyping of breast cancer into four subtypes according to PAM50 but the method is not restricted to this application.

**Strengths**:
- Extensive comparison to other fusion methods
- Good and sensible application for multimodal fusion methods

**Weaknesses**:
- Missing details on experimental evaluation: it is unclear whether there is a fixed test and how the best model is chosen on the validation set. Additionally, the experiments are performed in a 10-fold cross-validation setting but there is no standard deviation given. Finally, the chosen metrics are not suited for a multi-class classification with high class imbalance.
- Methods:
  - There is no major methodological difference between WL and WLB, this should rather be an ablation study and ideally the model should learn independently whether a non-zero bias is better or not.
  - WSI-image representations: SOTA pathology foundation models are not used as feature representations. There are many missing details in this subsection, e.g., how are the 50 most useful patches chosen?, which networks are ultimately used? why are there no slide representations used?
- Results:
  - Experimental results on the choice of the unimodal representations are missing.
  - State-of-the-art weakly-supervised baselines, such as attention-based multiple instance learning with patch-level foundation models or slide encoders, should be included to assess whether the unimodal baselines are acceptable.
  - Experiments with the option of WSI-graph without WSI-image would be interesting to see because WSI-graph performs better than WSI-image and the combination of WSI-image and WSI-graph is worse than only using WSI-graph.
  - There are no statistical test confirming the repeated claim of “significant” results and as mentioned above there are no standard deviations given to assess whether a max improvement of 2% is good or not.
- Code and detailed dataset information are missing to reproduce the results.

**Minor issues**:
- Fig. 1: unify naming of sub-panes (A, B, … vs (a), (b), …) or better: combine (i) and (ii) into one figure.
- Eq. 1: comma after equation
- p.4 bottom: reference to Fig 1i is wrong.
- Tables: unnecessary to have two symbols for best performance (bold and †)
- Table 2 and 3: optimize space by not repeating all abbreviations including the citations. It is sufficient to cite the relevant work in the main text.
- Table 3, last row: three best results with same value, which should all be highlighted.
- Why do you make a conceptual difference between accuracy and AUROC by showing accuracy in percentage and AUROC in floating point numbers? This refers to the text as well as to the tables.
- p. 8: Missing brackets around one reference

---

### Official Review · Reviewer_uyCt · 2025-07-14
**Good paper, fits well to the workshop**

**Rating:** 5
**Confidence:** 4

**Review:**

Paper Summary:
The authors propose a multimodal pipeline that is flexible for adding additional
modalities. They apply and evaluate it on CNV, EHR and WSIs as modalities for breast
cancer subtyping.
To further improve performance, they propose a new late fusion strategy and additionally
use a dual-representation of WSIs (image- and graph based). In their joint evaluation on
PAM50 breast cancer subtyping on data from TCGA.BRCA, TCGA and cBioPortal they
show an improved classification performance compared to comparison methods such
as Weighted Ensemble , Simple Ensemble Max-Model Predictor, Meta-Learning and
Intermediate Fusion.

Strengths:
* Interesting approach of combining image- and graph based representation of
WSIs. The results that indicate the improvement from this combination show the
significant potential of this method that could possibly even be applied to other
image modalities.

* Addressing an important problem of combining these- and potentially additional
medical modalities with an improved late fusion method that shows significant
performance improvement in the evaluation.

* Detailed evaluation with a large selection of comparison methods that plausibly
breaks down to each modality as presented in table 1

* Detailed analysis of stratification that indicates improvements from the proposed
weighted logits fusion

Weaknesses:
* This work states a flexibility of the pipeline to additional modalities. It remains
unclear, how extendable it actually is-especially as there is no code provided.

* I would expect a significantly increased requirement for computational resources
from the dual representation of the WSIs. Due to the improved classification
performance, it might still be worth the additional computational effort for
training/inference but it would help to have an evaluation of the impact on
computational resources to quantify the computational cost.

Justification:
This work introduces multiple novel concepts for multimodal breast cancer subtyping
that’s combination significantly outperforms existing comparison methods. The stated
improved flexibility from the multimodal pipeline is not sufficiently proven-especially as
no code is provided. But due to the remaining significant novelties and improvements
that stem from the automatic dual representation of WSIs as image and graphics and
the improved multimodal fusion, the recommendation is to accept this work.